# Proteomic Identification of Markers of Membrane Repair, Regeneration and Fibrosis in the Aged and Dystrophic Diaphragm

**DOI:** 10.3390/life12111679

**Published:** 2022-10-22

**Authors:** Stephen Gargan, Paul Dowling, Margit Zweyer, Michael Henry, Paula Meleady, Dieter Swandulla, Kay Ohlendieck

**Affiliations:** 1Department of Biology, Maynooth University, National University of Ireland, W23 F2H6 Maynooth, Ireland; 2Kathleen Lonsdale Institute for Human Health Research, Maynooth University, W23 F2H6 Maynooth, Ireland; 3Department of Neonatology and Paediatric Intensive Care, Children’s Hospital, German Center for Neurodegenerative Diseases, University of Bonn, D53127 Bonn, Germany; 4National Institute for Cellular Biotechnology, Dublin City University, D09 E432 Dublin, Ireland; 5Institute of Physiology, University of Bonn, D53115 Bonn, Germany

**Keywords:** annexin, biomarker, collagen, degeneration, dystrophinopathy, fibrosis, membrane repair, muscular dystrophy, periostin, regeneration

## Abstract

Deficiency in the membrane cytoskeletal protein dystrophin is the underlying cause of the progressive muscle wasting disease named Duchenne muscular dystrophy. In order to detect novel disease marker candidates and confirm the complexity of the pathobiochemical signature of dystrophinopathy, mass spectrometric screening approaches represent ideal tools for comprehensive biomarker discovery studies. In this report, we describe the comparative proteomic analysis of young versus aged diaphragm muscles from wild type versus the dystrophic *mdx-4cv* mouse model of X-linked muscular dystrophy. The survey confirmed the drastic reduction of the dystrophin-glycoprotein complex in the *mdx-4cv* diaphragm muscle and concomitant age-dependent changes in key markers of muscular dystrophy, including proteins involved in cytoskeletal organization, metabolite transportation, the cellular stress response and excitation-contraction coupling. Importantly, proteomic markers of the regulation of membrane repair, tissue regeneration and reactive myofibrosis were detected by mass spectrometry and changes in key proteins were confirmed by immunoblotting. Potential disease marker candidates include various isoforms of annexin, the matricellular protein periostin and a large number of collagens. Alterations in these proteoforms can be useful to evaluate adaptive, compensatory and pathobiochemical changes in the intracellular cytoskeleton, myofiber membrane integrity and the extracellular matrix in dystrophin-deficient skeletal muscle tissues.

## 1. Introduction

Muscular dystrophies form a large group of progressive muscle wasting diseases in the domain of inherited neuromuscular disorders [1,2,3] and are triggered by a variety of primary abnormalities [4]. Mutations in the *DMD* gene cause dystrophinopathies, including severe Duchenne muscular dystrophy of early childhood and its more benign and later onset form named Becker’s muscular dystrophy [5,6]. Dystrophinopathies are characterized by the almost complete loss or abnormal size of the dystrophin protein isoform Dp427-M in skeletal muscles [7,8]. The *DMD* gene is extremely large and contains several promoters [9]. Dystrophin proteins of 45 kDa to 427 kDa are expressed in a tissue-specific pattern [10]. Full-length isoforms of dystrophin are mostly found in muscular and neuronal tissues [11,12,13] with major cytoskeletal functions [14,15].

In skeletal muscles, dystrophin acts as a molecular anchor of a sarcolemmal glycoprotein complex that stabilizes the cellular periphery by indirectly linking extracellular merosin (laminin-211) of the basal lamina to cortical γ-actin of the intracellular cytoskeleton [16,17,18]. The connection of the full-length dystrophin isoform to the microtubular network makes the Dp427-M protein a member of the family of giant cytolinkers in skeletal muscles [19]. The dystrophin complex also supports cellular signaling pathways and provides lateral force transmission at costamers [20]. The members of the core dystrophin-glycoprotein complex include the Dp427-M isoform of dystrophin, α/β-dystroglycans, α/β/γ/δ-sarcoglycans, sarcospan, α-dystrobrevin and α/β-syntrophins [21,22].

Importantly, the expression of the dystrophin-glycoprotein complex is greatly reduced in dystrophinopathy [23,24,25] resulting in the loss of trans-plasmalemmal linkage in dystrophic skeletal muscles [26]. This was shown to impair sarcolemmal integrity and render dystrophin-deficient fibers more susceptible to membrane rupturing, which in turn triggers abnormal ion homeostasis and impaired signaling, affecting especially calcium handling and excitation-contraction coupling in muscular dystrophy [27,28,29]. Dystrophinopathies are mainly characterized by progressive skeletal muscle weakness, cognitive impairment in a subset of Duchenne patients, articular deformities, scoliosis and the occurrence of cardio-respiratory failure, which represent the main cause of death [6,30,31]. In addition to skeletal muscle weakness, Duchenne muscular dystrophy is associated with a variety of less severe and body-wide complications affecting liver metabolism, the gastrointestinal tract, the immune system and kidney function [30,31,32]. This makes X-linked muscular dystrophy a multi-systems disease with a primary neuromuscular pathology, which is reflected by complex changes in the serum biomarker profile of Duchenne patients [33,34,35].

In order to improve the differential diagnosis, prognostic evaluation and therapeutic-monitoring of the complex pathophysiology of muscular dystrophy, a considerable number of systematic studies have been applied to screen both tissue and biofluid specimens [36,37,38]. Proteomic searches for novel biomarker candidates have included clinical samples from patients afflicted with Duchenne/Becker muscular dystrophy [39,40] and various animal models of dystrophinopathy, such as dystrophic mice, dogs and pigs [41,42,43,44,45,46]. Animal models play a key role in the initial identification of new biomarkers [47,48,49,50] that can then be utilized for the evaluation of novel therapeutic approaches to treat dystrophinopathies [51,52,53,54]. Proteome-wide changes in the skeletal musculature were established to occur in the contractile apparatus, signaling pathways, the cytoskeletal network, metabolite transportation, bioenergetic pathways, the cellular stress response and the extracellular matrix [29,55,56,57]. Building on these findings, the new study presented in this report aimed at studying the effects of aging on dystrophinopathy and establish a comprehensive biomarker signature of the advanced stages of X-linked muscular dystrophy.

The mass spectrometry-based proteomic screening of young versus aged *mdx-4cv* diaphragm muscle described in this report has revealed complex proteome-wide changes in relation to dystrophin deficiency. Especially interesting was the identification of drastic and age-related increases in crucial markers of membrane repair regulation, tissue regeneration and reactive myofibrosis, i.e., annexins [58,59,60], dysferlin [61,62,63], caveolins [64,65,66], integrins [67,68,69], cadherin [70,71,72], CD34 [73,74,75], periostin [76,77,78] and collagens [79,80,81]. Changes in annexin 2, periostin and collagen VI were independently confirmed by comparative immunoblot analysis.

This is of considerable biomedical interest since annexins are intrinsically involved in the regulation of membrane repair [82,83,84], and the extracellular collagen network and its associated components play a crucial role in the stabilization, force transmission and functioning of skeletal muscles [85,86,87]. Considerable crosstalk occurs between the extracellular matrix and contractile fibers in various myopathies [88,89,90]. Thus, the elevated expression levels of these crucial markers of regeneration, membrane repair and reactive myofibrosis at advanced stages of X-linked muscular dystrophy could be useful for the establishment of robust and informative indicators of changes in the sarcolemma and extracellular matrix region during the progression of dystrophinopathy [91,92,93].

## 2. Materials and Methods

### 2.1. Materials

General materials and analytical reagents for the comparative proteomic survey of normal versus dystrophic diaphragm muscles were obtained from Bio-Rad Laboratories (Hemel-Hempstead, Hertfordshire, UK), GE Healthcare (Little Chalfont, Buckinghamshire, UK) and Sigma Chemical Company (Dorset, UK). Protease inhibitor cocktail tablets were from Roche (Mannheim, Germany). For digestion of protein samples, MS-grade trypsin protease was purchased from ThermoFisher Scientific (Dublin, Ireland). Filter-aided sample preparations were carried out with Vivacon 500 spin filters (VN0H22; 30,000 MWCO) from Sartorius (Göttingen, Germany). Precast Invitrogen Bolt 4–12% Bis-Tris gels and Whatman nitrocellulose transfer membranes were used for one-dimensional gel electrophoresis and immunoblotting, respectively, and obtained from Bio-Science Ltd. (Dun Laoghaire, Ireland). For the visualization of gel-separated protein bands, InstantBlue Coomassie Protein Stain was purchased from Expedeon (Heidelberg, Germany). Primary antibodies for immunoblotting were from Invitrogen, Waltham, MA, USA (mAb SD83-03 against collagen VI), Abcam, Cambridge, UK (pAb ab41803 to annexin ANXA2) and Novus Biologicals, Cambridge, UK (mAb NBP1-30042 against periostin). Secondary peroxidase-conjugated anti-IgG were purchased from Sigma Chemical Company (Dorset, UK). Chemiluminescence kits were from Roche (Mannheim, Germany). The Pierce 660 nm Protein Assay Reagent was purchased from ThermoFisher Scientific (Dublin, Ireland).

### 2.2. Wild Type and Dystrophic Diaphragm Muscle Specimens

The dissection of freshly prepared *postmortem* diaphragm specimens from 3-month and 15-month old wild type C57BL6 mice and aged-matched samples from the *mdx-4cv* mouse model of Duchenne muscular dystrophy was performed according to institutional regulations. Normal controls and dystrophic mice were handled in strict adherence to local governmental and institutional animal care regulations and were approved by the Institutional Animal Care and Use Committee (Amt für Umwelt, Verbraucherschutz und Lokale Agenda der Stadt Bonn, North Rhine-Westphalia, Germany). Frozen specimens were transported on dry ice to Maynooth University in accordance with the regulations of the Department of Agriculture (animal by-product register number 2016/16 to the Department of Biology, National University of Ireland, Maynooth). Comparative proteomic analyses were carried out with muscle samples from 6 wild type and 6 dystrophic mice. Verification analyses using immunoblotting were carried out with specimens derived from a minimum of 4 wild type and 4 dystrophic mice.

### 2.3. Preparation of Muscle Tissue Extracts

The preparation of tissue specimens and subsequent analysis of extracted proteins by bottom-up proteomics was performed by a standardized procedure, which has recently been described in a detailed methods paper [94]. Wild type C57BL6 and dystrophic *mdx-4cv* mice of differing age were sacrificed in the Bioresource Unit of the University of Bonn and diaphragm muscle specimens then quick-frozen in liquid nitrogen [45]. Frozen samples were transported on dry ice to Maynooth University and stored at −80 °C prior to comparative proteomic analysis. Both, young and aged diaphragm samples were homogenized in a comparative way using lysis buffer containing 4% (*w*/*v*) sodium dodecyl sulfate, 0.1 M dithiothreitol and 100 mM Tris-Cl, pH 7.6, as well as a protease inhibitor cocktail [95]. Homogenization was carried out with a handheld homogenizer (Kimble Chase, Rockwood, TN, USA) [96]. Following homogenization, samples were briefly treated in a sonicating water bath, then heated for 3 min at 95 °C and centrifuged at 16,000× *g* for 5 min. The protein-containing supernatant was extracted and used for subsequent proteomic studies [95]. The Pierce 660 nm Protein Assay system was used to determine protein concentration [97]. Extracted diaphragm proteins were further processed for mass spectrometric analysis. Samples were mixed with 200 μL of 8 M urea, 0.1 M Tris pH 8.9 in Vivacon 500 spin filter units and centrifuged at 14,000× *g* for 15 min. The filter-aided sample preparation (FASP) method, developed by Wiśniewski and co-workers [98], was used for sample processing, buffer switching and protein trypsination prior to mass spectrometric peptide analysis [94].

### 2.4. Mass Spectrometry and Proteomic Data Analysis

The comparative label-free liquid chromatography mass spectrometric analysis of young versus aged diaphragm muscles from wild type versus the dystrophic *mdx-4cv* mouse model of dystrophinopathy was performed with a Thermo Orbitrap Fusion Tribrid mass spectrometer from Thermo Fisher Scientific (Waltham, MA, USA). Details of the proteomic workflow describing all preparative steps and analytical procedures using data-dependent acquisition, as well as bioinformatic data handling, were recently outlined in detail [94]. Reverse-phased capillary high-pressure liquid chromatography was carried out with a Thermo UltiMate 3000 nano system and directly coupled in-line with the Thermo Orbitrap Fusion Tribrid mass spectrometer. The UniProtKB-SwissProt *Mus musculus* database with Proteome Discoverer 2.2 using Sequest HT (Thermo Fisher Scientific) and Percolator were employed for the qualitative data analysis of mass spectrometric files. For the identification of diaphragm proteins, the following crucial search parameters were employed: (i) a value of 0.02 Da for MS/MS mass tolerance, (ii) a value of 10 ppm for peptide mass tolerance, (iii) variable modification settings for methionine oxidation, (iv) fixed modification settings in relation to carbamido-methylation and (v) tolerance for the occurrence of up to two missed cleavages. Peptide probability was set to high confidence. A minimum XCorr score of 1.5 for 1, 2.0 for 2, 2.25 for 3 and 2.5 for 4 charge state was employed for the filtering of peptides. The software analysis programme Progenesis QI for Proteomics (version 2.0; Nonlinear Dynamics, a Waters company, Newcastle upon Tyne, UK) was used to carry out quantitative label-free data analysis. Proteome Discoverer 2.2 using Sequest HT (Thermo Fisher Scientific) and a percolator were employed for the identification of peptides and proteins. Datasets were imported into Progenesis QI software for further analysis. Following the review of protein identifications, only those data that agreed with a crucial set of criteria were deemed as differentially expressed protein species between experimental groups based on statistical significance and high confidence. The criteria included an ANOVA p-value of ≤0.01 between experimental groups, and proteins with ≥2 unique peptides contributing to the identification. The Progenesis QI programme calculated the mean abundance for individual protein species in each experimental condition to determine the maximum fold change for particular proteins. Condition-vs-condition matrixes with mean values were then used to determine the maximum fold change between any two condition’s mean protein abundances [94]. The multi-consensus MS files and listings of altered proteins in wild type versus dystrophic diaphragm muscle that were generated by the comparative proteomic study shown in this report have been deposited (15 September 2022) under the title ‘Proteomic analysis of aged *mdx-4cv* diaphragm’ with the unique identifier ‘qmtre’ to the Open Science Foundation (https://osf.io/qmtre/, accessed on 15 September 2022). The standard bioinformatic analysis tool PANTHER [99] was utilized for the identification of protein classes.

### 2.5. Comparative Immunoblot Analysis

Primary antibodies to the matricellular protein periostin (POSTN), the extracellular matrix component collagen (COL-VI) and the Ca^2+^-dependent membrane repair protein annexin (ANXA2) were used for the independent verification of key findings from the mass spectrometry-based proteomic screening of young versus aged *mdx-4cv* diaphragm muscle preparations. One-dimensional gel electrophoresis and immunoblot analysis were carried out by standardized methodology [95]. Diaphragm samples were incubated in Laemmli-type sample buffer and heated for 30 min at 37 °C. Per lane, 20 μg of protein were ran on Invitrogen Bolt 4–12% Bis-Tris gels. Coomassie staining of protein gels was performed with InstantBlue Coomassie Protein Stain. Gel electrophoretically separated proteins were transferred to nitrocellulose membranes for immunoblot analysis. Blocking was performed with a fat-free milk solution and membrane sheets were incubated overnight with 1:1000 (*v*/*v*) diluted primary antibodies. Detection of labelled protein bands was carried out by incubation with 1:1000 (*v*/*v*) diluted peroxidase-conjugated secondary antibodies and enhanced chemiluminescence. Statistical analysis of immunoblots (*n* = 4) was performed with ImageJ software (NIH, Bethesda, MD, USA), along with Microsoft Excel, in which statistical significance was based on a *p*-value ≤ 0.05.

## 3. Results

The comparative proteomic profiling of the dystrophin-deficient *mdx-4cv* diaphragm was carried out with a Thermo Orbitrap Fusion Tribrid mass spectrometer and multi-consensus analysis identified 2421 and 2790 protein species in wild type versus *mdx-4cv* preparations, respectively. Throughout the manuscript, names of proteins, protein subunits or protein isoforms that were identified by proteomics are abbreviated in capital letters. Differential protein expression patterns were analyzed by bioinformatics and are displayed in this report according to association with the (i) dystrophin-glycoprotein complex, (ii) the established marker signature of dystrophinopathy, (iii) excitation-contraction coupling, (iv) the annexin family, (v) the collagen family, and (vi) the extracellular matrix. The increased abundance of collagen COL-VI as marker of reactive myofibrosis, the multi-functional periostin (POSTN) as a prototype of a matricellular component and the membrane repair protein annexin ANXA2 were verified by immunoblotting.

### 3.1. Mass Spectrometric Analysis of Young versus Aged mdx-4cv Diaphragm Muscle

Overall changes in protein classes were determined by bioinformatic PANTHER analysis [99]. The findings are displayed in Figure 1 and Figure 2, which show drastic alterations in a variety of protein types, including RNA metabolism proteins, cytoskeletal proteins, metabolite interconversion enzymes, protein modifiers and translational proteins. An interesting change between 3-month and 15-month old *mdx-4cv* diaphragm muscle is the drastic increase in the changed density of metabolite interconversion enzymes during aging (Figure 2).

### 3.2. Reduced Dystrophin Complex in mdx-4cv Diaphragm Muscle

Following the proteomic analysis of young versus aged *mdx-4cv* diaphragm muscle, the mutant status of the analyzed dystrophic muscle specimens was confirmed by comparison between wild type and *mdx-4cv* samples. As diagrammatically shown in Figure 3, the dystrophin complex consists of the cytoskeletal Dp427-M isoform of dystrophin, integral β-dystroglycan, extracellular α-dystroglycan, α/β/γ/δ-sarcoglycans, sarcospan, α-dystrobrevin and α/β-syntrophins [20]. Linkage to this sarcolemma-associated complex occurs to extracellular collagen isoforms via laminin-211, and intracellularly dystrophin binds to γ-actin filaments and tubulins of the microtubular network.

Representative members of the core dystrophin-glycoprotein complex were identified by mass spectrometry, i.e., dystrophin (DMD), α/β-dystroglycan (DAG1), α-sarcoglycan (SGCA), β-sarcoglycan (SGCB), α-dystrobrevin (DTNA) and α1-syntrophin (SNTA1), and shown to be greatly reduced in both young and aged *mdx-4cv* diaphragm (Figure 3). The most drastic change between young and old muscle specimens were observed for α-dystrobrevin in 15-month old *mdx-4cv* diaphragm muscle.

### 3.3. Established Changes of Dystrophic Biomarkers in mdx-4cv Diaphragm Muscle

In order to put this new study into perspective, a variety of established proteomic markers of X-linked muscular dystrophy were analyzed. As shown in Figure 4, increases in the intermediate filament component vimentin (VIM), the microtubular protein isoform tubulin-alpha-1c (TUBA1C), the intracellular iron regulatory protein ferritin light chain (FTL1), heat shock protein A5 (HSPA5), small heat shock protein B7 (HSPB7), large heat shock protein 90AA1 (HSP90AA1) and nuclear lamin isoform B2 (LMNB2) were confirmed. Characteristic decreases were established for the cytosolic calcium-signaling protein parvalbumin (PVALB) and the metalloenzyme carbonic anhydrase isoform 3 (CA3).

### 3.4. Proteomics of Excitation-Contraction Coupling and Calcium Handling in mdx-4cv Diaphragm

Since micro-rupturing of the dystrophin-deficient sarcolemma membrane causes disturbed calcium homeostasis and abnormal cellular signaling in X-linked muscular dystrophy, it was of interest to evaluate the status of proteins involved in excitation-contraction coupling and intracellular calcium handling. This included the mass spectrometric survey of the ryanodine receptor calcium-release channel (RYR1) of the sarcoplasmic reticulum, the transverse tubular L-type calcium channel (Caᵥ1.1) with its voltage-sensing α1S-subunit (CACNA1S) and the auxiliary subunits β1 (CACNB1) and α2/δ (CACNA2D1), the calcium-binding protein calsequestrin (CASQ1) of the terminal cisternae region, the fast SERCA1 isoform of the sarco(endo)plasmic reticulum calcium-ATPase (ATP2A1), and the junctophilins 1 (JPH1) and 2 (JPH2) of the triad junction (Figure 5).

The principal ion channel-containing α1S-subunit of the transverse tubular L-type calcium channel and its auxiliary β1-subunit were shown to be reduced in both young and aged dystrophic diaphragm muscle. In contrast, the ryanodine receptor, the α2/δ-subunit of the transverse tubular calcium channel, fast calsequestrin, the fast sarcoplasmic reticulum calcium ATPase SERCA1 and junctophilins exhibited a reduced expression level only in 15-month old *mdx-4cv* diaphragm.

### 3.5. Proteomics of Membrane Repair and Calcium Sensing in mdx-4cv Diaphragm

A protein of central importance in the calcium-dependent membrane repair process of damaged skeletal muscles is dysferlin (DYSF). The mass spectrometric survey of wild type versus *mdx-4cv* diaphragm preparations revealed an increase of this sarcolemmal repair protein. Since the dysferlin/myoferlin system closely interacts with caveolins and annexins, the expression levels of these crucial components of the caveolae structures and calcium regulation, respectively, were also examined. The concentration of caveolin 1 (CAV1) and caveolin 2 (CAV2) increased drastically in aged and dystrophic diaphragm, while the muscle-specific caveolin-3 (CAV3) isoform showed elevated levels in 3-month old *mdx-4cv* diaphragm but only marginal changes in 15-month old dystrophic muscles. Integrins are useful markers of proliferation and the main type present in the skeletal muscle periphery, α7β1-integrin (ITGA7/ITGB1), was identified to be increased in both young and aged *mdx-4cv* diaphragm. Interestingly, the expression of the myogenic markers cadherin-13 (CDH13) and CD34 were both elevated in muscular dystrophy.

The calcium-dependent annexin isoforms A1 to A7 (ANXA1, ANXA2, ANXA3, ANXA4, ANXA5, ANXA7) of which some are involved in calcium-related and dysferlin-associated repair mechanisms of the muscle surface membrane system, were analyzed in 3-month versus 15-month old *mdx-4cv* diaphragm muscle preparations. The degree of increased expression levels in young versus aged *mdx-4cv* diaphragm was found to be relatively comparable (Figure 6).

### 3.6. Proteomics of Collagens and the Extracellular Matrix in mdx-4cv Diaphragm Muscle

Skeletal muscles contain a considerable number of collagen isoforms and its extracellular matrix is formed by a highly complex mesh of diverse proteins. The comparative proteomic analysis of 3-month versus 15-month old *mdx-4cv* diaphragm muscle revealed increases in collagen I (COL1A1 and COL1A2 chains), collagen IV (COL4A1, COL4A2 and COL4A3 chains), collagen V (COL5A1 and COL5A2 chains), collagen VI (COL6A1, COL6A2, COL6A5 and COL6A6 chains), collagen XII (COL12A1 chain), collagen XIV (COL14A chain), collagen XV (COL15A1 chain) and collagen XVIII (COL18A1 chain), as shown in Figure 7. Increased levels of crucial extracellular matrix proteins were identified in the case of the matricellular protein periostin (POSTN), the small leucine-rich proteoglycans asporin (ASPN), biglycan (BGN), lumican (LUM) and mimecan/osteoglycin (OGN), the adhesive glycoprotein fibronectin (FN1), the proteoglycan decorin (DCN), the tyrosine-rich acidic matrix protein dermatopontin (DPT), the hemopexin-type glycoprotein vitronectin (VTN) and the basal lamina component nidogen-2/osteonidogen (NID2). The most drastic differences in the degree of increased expression levels between young versus aged *mdx-4cv* diaphragm were found for the collagen chains COL5A1, COL5A2, COL6A5 and COL-XIV, as well as the matricellular protein periostin (Figure 7).

### 3.7. Immunoblotting of Collagen VI, Periostin and Annexin 2 in mdx-4cv Diaphragm

Trends in changed protein expression pattern as determined by the above-described comparative mass spectrometric investigation of protein extracts from 3-month versus 15-month old *mdx-4cv* diaphragm muscle were independently verified by immunoblotting, as shown in Figure 8.

Coomassie Blue staining of one-dimensionally separated diaphragm proteins revealed no major differences in the band pattern of 3-month versus 15-month old wild type versus *mdx-4cv* diaphragm preparations. However, immunoblotting with antibodies to collagen COL-VI, periostin and annexin isoform ANXA2 clearly showed a significant increase of these proteins in dystrophic diaphragm muscle. Both, collagen and periostin exhibited high expression levels in aged and dystrophin-deficient muscles (Figure 8).

## 4. Discussion

The comparative proteomic analysis described in this report was carried out with young versus aged diaphragm muscles from wild type versus the dystrophic *mdx-4cv* mouse model of Duchenne muscular dystrophy. The *mdx*-type models of Duchenne muscular dystrophy are based on the spontaneous *mdx-23* mouse [100] which is characterized by a point mutation in exon 23 of the *DMD* gene [101]. This results in the almost complete deficiency of dystrophin isoform Dp427-M [47] and causes an X-linked myopathy in association with muscular hypotrophy, hypertrophy and hyperplasia in *mdx-23* fibres [102]. Especially the diaphragm muscle is severely affected in *mdx*-type mice and exhibits progressive fiber degeneration and reactive myofibrosis [103,104,105] making the dystrophin-deficient diaphragm a suitable tissue for studying the molecular and cellular pathogenesis of Duchenne muscular dystrophy [47,49]. An alternative model of dystrophinopathy is the *mdx-4cv* mouse [106] that has been generated by chemical mutagenesis with N-ethylnitrosourea which induced a C-to-T transition at position 7916 in exon 53 [107]. This nonsense point mutation results in the premature termination of translation and the production of a truncated and non-functional dystrophin protein [108]. Importantly, skeletal muscles from the *mdx-4cv* mouse display approximately 10-fold fewer revertant and dystrophin-positive contractile fibers as compared to the *mdx-23* mouse [109]. The drastically reduced presence of revertant fibers makes the *mdx-4cv* mouse an attractive model and has therefore been widely used for (i) the detailed analysis of dystrophic changes in the skeletal musculature [95,96,110,111,112], (ii) studying dystrophinopathy-related cardiomyopathy [113], (iii) evaluating the protective phenotype of extraocular muscle [114], (iv) surveying of bodily fluids for biomarker candidates including serum, urine and saliva [115,116,117], (v) the screening of body-wide changes in the multi-systems pathology of X-linked muscular dystrophy including proteome-wide alterations in liver, spleen, kidney, stomach and brain [118,119,120,121,122] and (vi) the testing of new therapeutic procedures to treat muscular dystrophy [123,124,125].

Building on these studies, we have used the aged *mdx-4cv* diaphragm muscle as a model system to study the advanced stages of X-linked muscular dystrophy. Crucial findings from the mass spectrometric survey include the detection of increased levels of key marker proteins that are involved in the regulation of membrane repair, tissue regeneration and reactive myofibrosis. Annexin isoform ANXA2 [58,59,60], the matricellular protein periostin [76,77,78] and collagen isoform COL-VI [79,80,81], as well as the caveolins CAV1 and CAV2 [64,65,66], α7β1-integrin [67,68,69], cadherin-13/T-cadherin [70,71,72] and CD34 [73,74,75] were shown to be drastically increased in aged and dystrophin-lacking muscle preparations. Interestingly, the muscle-specific caveolin isoform CAV3 was only elevated in young and dystrophic diaphragm. Immunoblotting clearly verified the increased concentration of specific isoforms of annexin, periostin and collagen in muscular dystrophy. This suggests that these markers of membrane repair, tissue regeneration and myofibrosis are suitable to characterize muscle biopsy specimens from genetic animal models of dystrophinopathy. Since the spontaneous *mdx-23* mouse is the most frequently used animal model in muscular dystrophy research [47], it was of interest to compare the proteomic changes in the chemically induced *mdx-4cv* mouse [106,107,108] to the naturally occurring *mdx-23* mutant [100,101,102]. The proteome-wide changes identified in this report on established markers of X-linked muscular dystrophy, i.e., decreases in all members of the core dystrophin complex and concomitant increases in vimentin, tubulin, ferritin, various molecular chaperones and lamin, and decreases in parvalbumin and carbonic anhydrase isoform CA3, agree with previous studies of the dystrophic *mdx-23* mouse model [41,42,43,44,45,46,126,127]. An increased abundance of collagen COL-VI was also observed in *mdx-23* muscle preparations using two-dimensional gel electrophoresis combined with staining by the fluorescent dye ruthenium II tris bathophenanthroline disulfonate [128].

New biomarker candidates can now be used to improve diagnostic procedures, the accuracy of prognosis, therapeutic-monitoring and the evaluation of potential side effects due to novel pharmacological treatments or new gene therapeutic approaches, such as gene substitution, exon-skipping or gene editing [51,52,53,54]. Proteomic biomarkers can be utilized in a variety of crucial biomedical and clinical applications and be useful to improve the evaluation of susceptibility to muscular dystrophy, differential diagnosis, prognostics, prediction of patient sensitivity, pharmacodynamics, monitoring of therapeutic success and drug safety [35]. Figure 9 summarizes some of the key findings of this proteomic survey of aged and dystrophin-deficient *mdx-4cv* diaphragm muscle.

In relation to bioenergetic and metabolic enzymes, the bioinformatic PANTHER analysis of changed proteins revealed an extensive increase in the altered abundance of metabolite interconversion enzymes during aging of the dystrophic *mdx-4cv* diaphragm muscle. These considerable changes in the expression levels of muscle-associated enzymes included large numbers of hydrolases, isomerases, ligases, lyases, oxidoreductases and transferases. Within the protein class of hydrolases, this included proteins with amylase, deaminase, esterase, glucosidase, lipase, phosphatase, phosphodiesterase and pyrophosphatase activity. Isomerases were represented by epimerases, racemases and mutases. Lyases included aldolases, cyclases, dehydratases and hydratases. Alterations in the abundance of oxidoreductases encompassed dehydrogenases, oxidases, oxygenases, peroxidases and reductases. The protein family of transferases included acetyltransferases, acyltransferases, glycosyltransferases, kinases, methyltransferases and transketolases. A comprehensive biomarker signature of these types of tissue-associated changes in combination with relevant alterations in biofluids, such as saliva, urine and serum/plasma, can be extremely helpful to advance the field of muscular dystrophy research [33,34,35,36,37] and (i) determine the risk for disease prior to the appearance of initial symptoms, (ii) detect the differential nature of specific and potentially sequential pathophysiological processes including progressive fiber degeneration, impaired excitation-contraction coupling, abnormal ion homeostasis, dysregulated cellular signaling, chronic inflammation with immune cell invasion, fat substitution and reactive myofibrosis [28,29,30,31,32,91,92,93], (iii) evaluate disease progression and potential adverse events during treatment, (iv) determine individual sensitivities towards drug treatment, (v) reflect the metabolic/biotransformation responses of detoxifying organs such as the liver and kidneys following exposure to a novel therapeutic agent, (vi) monitor (ideally repeatedly) alterations in the disease status due to therapeutic interventions [51,52,53,54], and (vii) assure the absence of cytotoxic side effects on whole-body physiology [37].

This gives proteomic markers of calcium-dependent membrane repair, tissue regeneration and myofibrosis, as identified in this study, great potential for future usage in disease evaluation and therapeutic-monitoring. The drastic changes in the collagen isoforms COL-V, COL-VI and COL-IV [79] in aged *mdx-4cv* diaphragm identify these proteins as suitable markers of reactive myofibrosis [91]. Skeletal muscles contain a variety of collagens ranging from COL-I to COL-XXII [79,86]. The comparative proteomic survey of the dystrophic diaphragm identified collagens COL-I, COL-IV, COL-V, COL-VI, COL-XII, COL-XIV, COL-XIV and COL-XVIII, which are located to differing degrees in the basal lamina, endomysium, perimysium, epimysium and myotendinous junctions [85,86,87]. Collagen COL-IV is the basic structural component of the basal lamina and characterized by a helical form [86]. COL-IV functions as the main linker to the dystrophin-glycoprotein complex via laminin-211 and the α/β-dystroglycan subcomplex [16,20,21]. Collagen COL-V forms fibrils and is mostly found in the endomysium where it controls the process of collagen fibrillogenesis [86]. The beaded filaments of collagen COL-VI interact with various cell surface receptors and are intrinsically involved in the maintenance of skeletal muscle integrity [80,81]. COL-XIV links fibrillar collagens to other components of the extracellular matrix and is mostly located in the endomysium and perimysium [87]. The increased levels of these collagens play a crucial role in reactive myofibrosis and are the underlying cause for the loss of tissue elasticity in X-linked muscular dystrophy [91,92,93]. In relation to the progressive nature of X-linked muscular dystrophy, myofibrosis is an excellent indicator of overall muscle deterioration and correlates well with loss in motor functions in Duchenne patients [88,89].

Thus, protein markers of myofibrosis, such as collagen isoform COL-VI, in conjunction with elevated expression levels of other extracellular matrix components, represent excellent indicators of the progression of muscular dystrophy. Additional suitable proteins of the matrisome are the small leucine-rich proteoglycans asporin, biglycan, lumican and mimecan, as well as the proteoglycan decorin, fibronectin, dermatopontin, vitronectin and nidogen-2. The identified increased levels of the tyrosine-rich acidic matrix protein dermatopontin, the dystrophin-associated protein biglycan and the adhesive glycoprotein fibronectin in dystrophic skeletal muscles agree with previous proteomic studies using fluorescence two-dimensional difference in-gel electrophoresis [112,113]. Periostin is a crucial matricellular protein involved in tissue regeneration and cellular signaling events [76,78]. Previous studies indicate that periostin is only temporally expressed in the extracellular matrix during differentiation and regenerative processes [77], which agrees with the findings of the mass spectrometric study presented in this report. Expression levels of periostin are clearly affected by dystrophinopathy-associated changes in the extracellular matrix [91,93,129]. Notably, the deletion of periostin was shown to have a positive effect on X-linked muscular dystrophy by reducing myofibrosis via modulation of the signaling pathway that is associated with transforming growth factor TFG-β [130]. Immunoblotting indicates that periostin exists only at very low levels in normal wild type diaphragm. This makes this component of the extracellular matrix an excellent candidate for evaluating progressive alterations in dystrophin-deficient skeletal muscles.

The proteomic evaluation of key proteins involved in the regulation of calcium homeostasis has shown elevated levels of calcium-sensing annexins and a decrease in regulatory components of excitation-contraction coupling. The calcium hypothesis of Duchenne muscular dystrophy assumes that abnormal calcium handling plays a major role in progressive myonecrosis [27]. Full-length dystrophin forms a lattice underneath the sarcolemma in normal muscles, which stabilizes the fiber periphery by linking the actin cytoskeleton to the extracellular matrix via the dystrophin-associated dystroglycan complex [18,20]. The almost complete loss of the Dp427-M isoform in dystrophinopathy weakens this trans-plasmalemmal linkage and renders dystrophic fibers more susceptible to membrane leakage during excitation-contraction-relaxation cycles [28]. The resulting influx of calcium ions, both through membrane tears and calcium leak channels, causes a drastic elevation of the sarcosolic calcium concentration. This in turn activates calcium-dependent proteolysis, which results in the enhanced destruction of muscle proteins [27]. The reduced levels of the voltage-sensing L-type calcium channel of the transverse tubules, the ryanodine receptor calcium release channel and associated junctophilins of the triad junction, the luminal calcium-binding protein calsequestrin and the calcium-pumping ATPase of the sarcoplasmic reticulum agree with the pathophysiological concept of dysregulated calcium fluxes in dystrophin-deficient fibers [28].

The loss of key proteins involved in the temporal and spatial regulation of calcium movements through muscle membranes appears to be the underlying cause for impaired excitation-contraction coupling, decreased calcium buffering and abnormal calcium re-uptake into the lumen of the sarcoplasmic reticulum. These changes in ion-regulatory proteins might represent accumulating abnormalities and/or compensatory mechanisms in dystrophic diaphragm fibers [29]. Another interesting detection of altered expression levels is the mass spectrometric identification of increased levels of calcium-sensing annexins. This large family of proteins is involved in membrane repair [58] and annexin isoform ANXA2 was previously shown to be involved in myofiber repair in conjunction with inflammation and adipogenic replacement in injured contractile tissues [59]. Annexins are crucial facilitators of the accumulation of dysferlin, which is responsible for muscle fiber membrane repair [58,59,60,61,62]. Dysferlin, myoferlin and caveolins closely interact in dystrophic fibres [45,111]. This makes both the ANXA2 isoform of muscle annexin and elevated levels of dysferlin in damaged and dystrophin-deficient fibers [112,131] promising biomarker candidates of calcium-dependent and dysferlin/myoferlin-associated repair processes in X-linked muscular dystrophy.

Another interesting proteomic observation was the increase in integrins [67,68,69]. The α7β1-integrin complex is developmentally regulated and important for sarcolemmal stability and prevention of exercise-induced injury [132] and increased levels of α7β1-integrin were previously shown to counteract muscle degeneration [133,134,135]. Integrins play a key role in the reinforcement of crucial load-bearing structures at costamers and myotendious junctions. Together with the dystrophin-complex, integrins provide structural integrity for lateral and longitudinal force transmission across the sarcolemma [136]. Thus, the increased expression of α7β1-integrin might be part of the repair response in muscular dystrophy and represents a potential marker of cellular proliferation. Since this study was carried out with total extracts from crude muscle preparations, low-abundance markers were not detected. However, increases in the myogenic marker molecules CD34 [73,74,75] and cadherin-13 [71,72,137] were identified in the dystrophic *mdx-4cv* diaphragm by mass spectrometric analysis. The surface marker CD34 was recently shown to exhibit considerable potential as a satellite cell-linked biomarker of skeletal muscle aging [75]. Cadherin-13 (CDH13), also named heart cadherin (H-cadherin) or truncated cadherin (T-cadherin), attaches to the plasma membrane via a glycosylphosphatidylinositol anchor. Importantly, T-cadherin interacts with integrin-α7 and is linked to signal transduction proteins within caveolae structures [70]. This makes the identified alterations in the abundance of specific isoforms of integrin, cadherin and caveolin in muscular dystrophy an interesting finding with potential for the establishment of a comprehensive biomarker signature of dystrophinopathy.

## 5. Conclusions

The mass spectrometry-based proteomic survey of 3-month versus 15-month old diaphragm muscles from wild type versus the dystrophic *mdx-4cv* mouse model of Duchenne muscular dystrophy has identified meaningful biomarker candidates of membrane repair, tissue regeneration and reactive myofibrosis. This included calcium-sensing annexins and caveolins that are involved in dysferlin-related membrane repair and the matricellular protein periostin which plays a crucial role in the extracellular matrix, as well as various isoforms of collagen that indicate the progression of fibrotic changes in the basal lamina, endomysium, perimysium, epimysium and myotendinous junctions of dystrophic skeletal muscles. Although not all of these proteomic candidate markers may be suitable as robust surrogate biomarkers that precisely correlate with a realistic clinical endpoint in therapeutic trials, they can nevertheless be useful for determining the complex pathogenesis of dystrophic muscles and might also be useful to evaluate the varied effects of novel treatments [138]. Biomarker signatures that have been established by omics-type screening processes, and properly verified for their effectiveness and reliability by patient-related analyses, should be able to reflect the pathobiochemical complexity of monogenetic disorders, such as Duchenne muscular dystrophy [57]. In the future, biomarker-guided diagnostics and therapeutic-monitoring will probably play a more central role in pre-clinical studies with animal disease models and patient screening during the main phases of clinical trials.

## Figures and Tables

**Figure 1 life-12-01679-f001:**
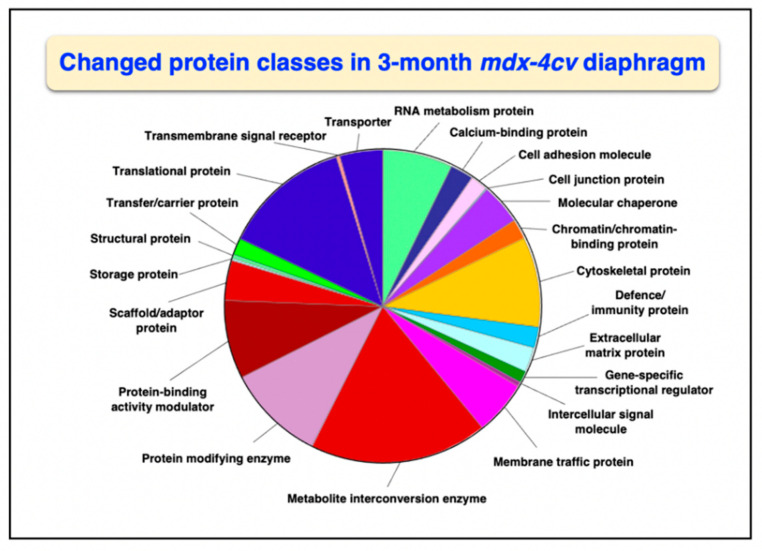
Bioinformatic analysis of changed protein classes in 3−month old *mdx-4cv* diaphragm muscle. The analysis was carried out with the PANTHER program [99].

**Figure 2 life-12-01679-f002:**
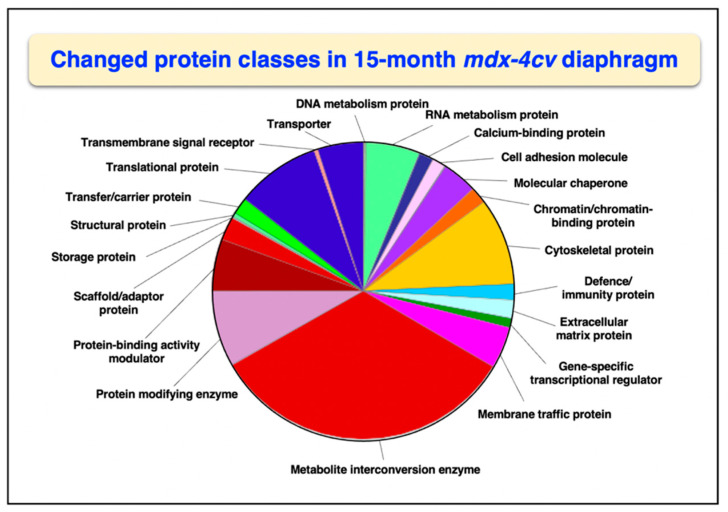
Bioinformatic analysis of changed protein classes in 15−month old *mdx-4cv* diaphragm muscle. The analysis was carried out with the PANTHER program [99].

**Figure 3 life-12-01679-f003:**
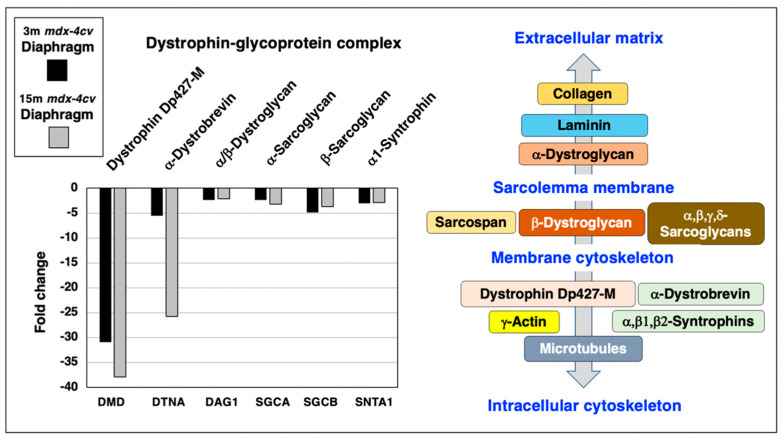
Proteomic analysis of the expression of the dystrophin-glycoprotein complex in 3−month (3 m, black bars) versus 15−month (15 m; grey bars) old *mdx-4cv* diaphragm muscle. On the left is shown a bar diagram of the fold change in abundance of representative members of the dystrophin-glycoprotein complex, as determined by comparative proteomics. Identified proteins included Dystrophin (DMD), α/β-Dystroglycan (DAG1), α-Sarcoglycan (SGCA), β-Sarcoglycan (SGCB), α-Dystrobrevin (DTNA) and α1-Syntrophin (SNTA1). On the right side is shown a diagrammatic presentation of the dystrophin-glycoprotein complex [20].

**Figure 4 life-12-01679-f004:**
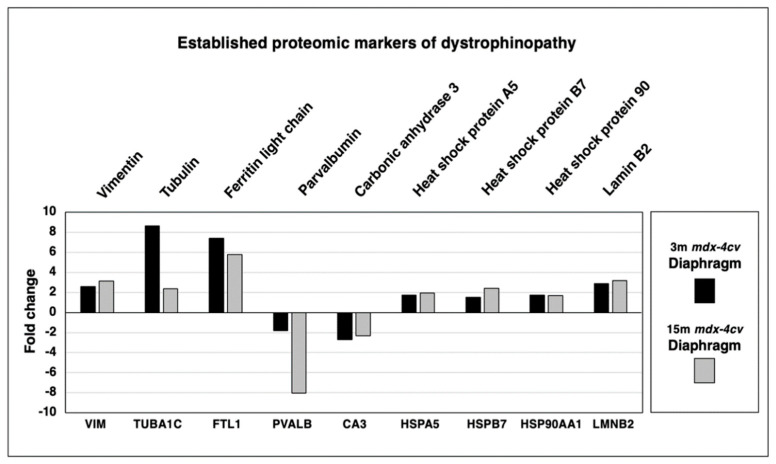
Proteomic analysis of the expression of established proteomic markers of X-linked muscular dystrophy in 3−month (3 m, black bars) versus 15−month (15 m; grey bars) old *mdx-4cv* diaphragm muscle. Identified proteins included Vimentin (VIM), Tubulin-alpha-1c (TUBA1C), Ferritin light chain (FTL1), Heat shock protein A5 (HSPA5), Heat shock protein B7 (HSPB7), Heat shock protein 90AA1 (HSP90AA1), Lamin isoform B2 (LMNB2), Parvalbumin (PVALB) and Carbonic anhydrase isoform 3 (CA3).

**Figure 5 life-12-01679-f005:**
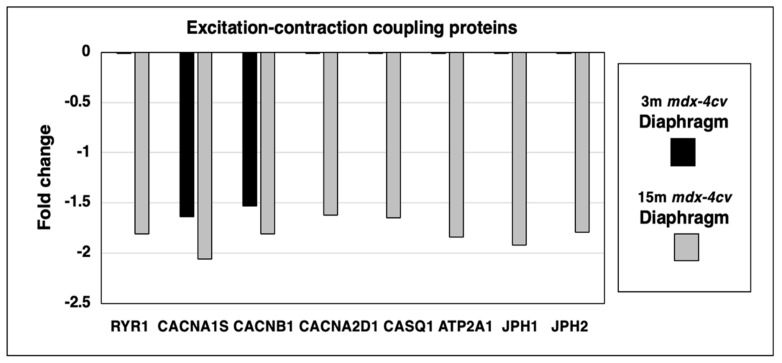
Proteomic identification of changes in the expression levels of key proteins involved in excitation-contraction coupling, calcium homeostasis and triad integrity in 3−month (3 m, black bars) versus 15−month (15 m; grey bars) old *mdx-4cv* diaphragm muscle. This included the Ryanodine receptor calcium-release channel (RYR1), the voltage-sensing L-type calcium channel with its α1S-subunit (CACNA1S), β1-subunit (CACNB1) and α2/δ-subunits (CACNA2D1), fast Calsequestrin (CASQ1), the fast SERCA1 Calcium-ATPase (ATP2A1), Junctophilin 1 (JPH1) and Junctophilin 2 (JPH2).

**Figure 6 life-12-01679-f006:**
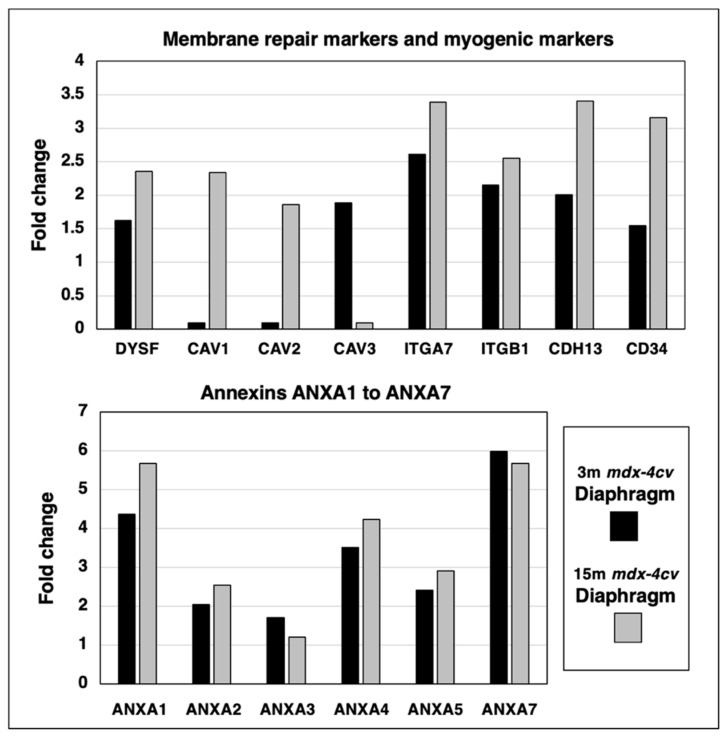
Proteomic identification of changes in the expression levels of key proteins involved in membrane repair and calcium sensing in 3−month (3 m, black bars) versus 15−month (15 m; grey bars) old *mdx-4cv* diaphragm muscle. This included the sarcolemmal membrane repair protein Dysferlin (DYSF), Caveolin 1 (CAV1), Caveolin 2 (CAV2), muscle-specific Caveolin 3 (CAV3), Integrin alpha-7 (ITGA7), Integrin beta-1 (ITGB1), Cadherin-13/T-cadherin (CDH13) and the myogenic marker CD34, as well as Annexin isoforms A1 (ANXA1), A2 (ANXA2), A3 (ANXA3), A4 (ANXA4), A5 (ANXA5) and A7 (ANXA7).

**Figure 7 life-12-01679-f007:**
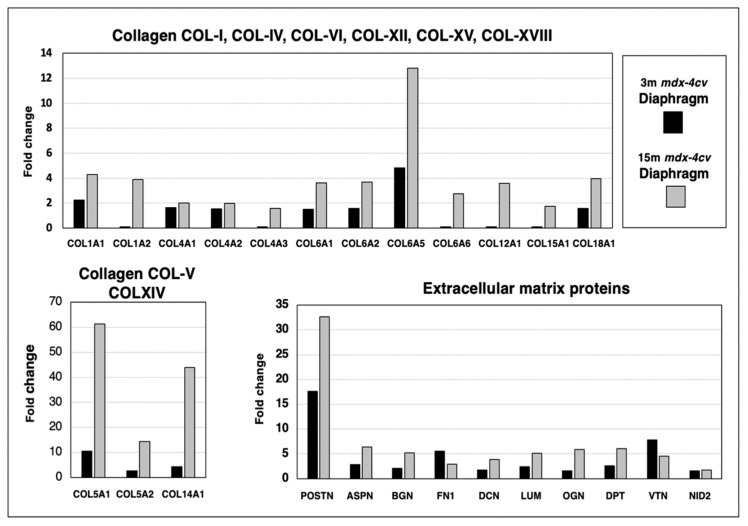
Proteomic identification of changes in the expression levels of collagens and related extracellular matrix proteins in 3−month (3 m, black bars) versus 15−month (15 m; grey bars) old *mdx-4cv* diaphragm muscle. This included Collagen I (COL1A1, COL1A2), Collagen IV (COL4A1, COL4A2, COL4A3), Collagen V (COL5A1, COL5A2), Collagen VI (COL6A1, COL6A2, COL6A5, COL6A6), Collagen XII (COL12A1), Collagen XIV (COL14A), Collagen XV (COL15A1), Collagen XVIII (COL18A1), Periostin (POSTN), Asporin (ASPN), Biglycan (BGN), Lumican (LUM), Mimecan/osteoglycin (OGN), Fibronectin (FN1), Decorin (DCN), Dermatopontin (DPT), Vitronectin (VTN) and Nidogen-2/osteonidogen (NID2).

**Figure 8 life-12-01679-f008:**
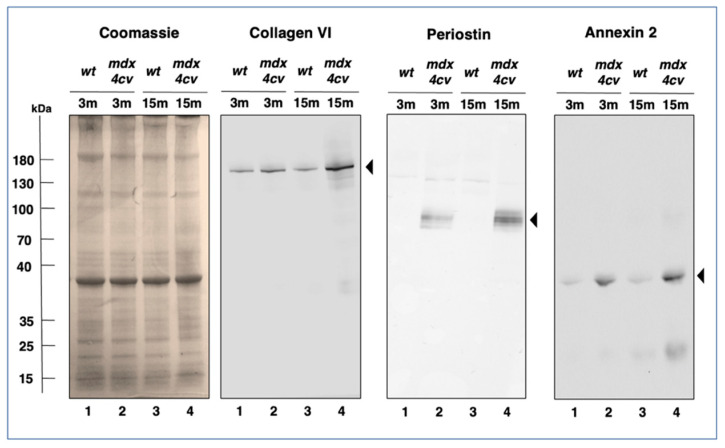
Immunoblot analysis of Collagen VI, Periostin and Annexin 2 in *mdx-4cv* diaphragm. Shown is a Coomassie-stained gel plus identical immunoblots labelled with antibodes to Collagen isoform COL-VI, the matricellular protein Periostin and Annexin isoform ANXA2. Lanes 1 to 4 contain protein extracts from 3−month (3 m) versus 15−month (15 m) old wild type versus *mdx-4cv* diaphragm, respectively. Molecular weight standards are marked on the left.

**Figure 9 life-12-01679-f009:**
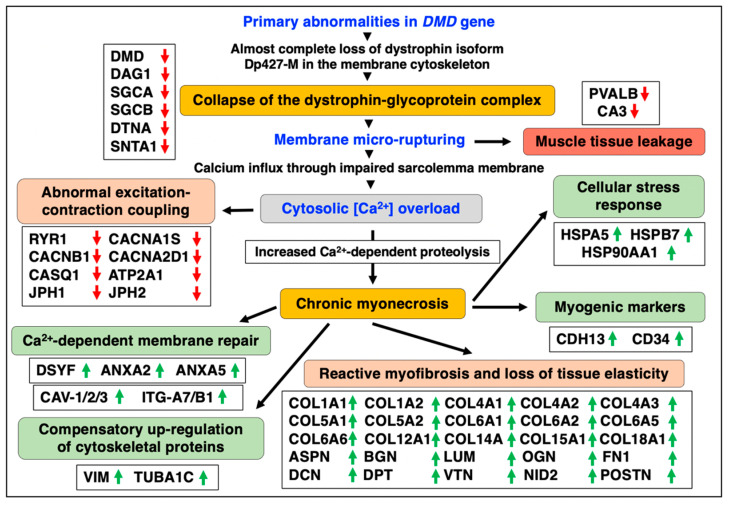
Overview of pathobiochemical and adaptive changes in the aged and dystrophin-deficient *mdx-4cv* diaphragm muscle as revealed by mass spectrometry-based proteomics. Listed are protein markers of crucial aspects of the molecular and cellular pathogenesis of dystrophinopathy, including the collapse of the dystrophin-glycoprotein complex, membrane rupturing, abnormal excitation-contraction coupling, reactive myofibrosis, calcium-dependent membrane repair, myogenic activation, the apparent compensatory up-regulation of cytoskeletal proteins, the cellular stress response and muscle tissue regeneration. Detected decreases in muscle-associated proteins are symbolized by red downward arrows and increases in proteins are marked by green upward arrows.

## Data Availability

The multi-consensus MS files and listings of altered proteins in wild type versus dystrophic diaphragm muscle that were generated by the comparative proteomic study shown in this report have been deposited under the title ‘Proteomic analysis of aged mdx-4cv diaphragm’ with the unique identifier ‘qmtre’ to the Open Science Foundation (https://osf.io/qmtre/) (accessed on 15 September 2022).

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
