# Peer review of "Proteomic Identification of Markers of Membrane Repair, Regeneration and Fibrosis in the Aged and Dystrophic Diaphragm"

_life, 2022, doi:10.3390/life12111679_

Round 1

Reviewer 1 Report

Identification of disease biomarkers can help evaluation of changes that result from the progressing pathology, are adaptive and compensatory, or result from the therapeutic intervention. Therefore, this study contributes to this goal. Proteomic identification of changes in the expression levels of proteins involved in the build-up of extracellular matrix and ultimately fibrosis is interesting and these have the potential to be used as biomarkers in pre-clinical and, upon confirmation in DMD patients, clinical therapeutic studies.

However, the study is focused entirely on the abnormalities in dystrophic myofibres and seems to be leaving aside changes that can occur in myogenic cells. Given that the progressive loss of the regenerative capacity of dystrophic muscles leaves ultimately to patients’ deaths, it would be interesting to re-evaluate the proteomic data for biomarkers of muscle repair.

In this context, periostin, albeit it has a variety of functions in tissues, in DMD can hardly be considered a tissue regeneration marker. Its ablation alleviated the dystrophic phenotype (Lorts A, Schwanekamp JA, Baudino TA, McNally EM, Molkentin JD. Deletion of periostin reduces muscular dystrophy and fibrosis in mice by modulating the transforming growth factor-β pathway. Proc Natl Acad Sci U S A. 2012 Jul 3;109(27):10978-83.)

Given that the mdx mouse is the most widely used pre-clinical model of DMD, it would be important to test the key protein alterations in this mouse to confirm that the same changes occur in this natural mutant as in this chemically-induced mdx4cv mice used here.

It is interesting that most of the alterations in the calcium-related proteins were identified in old diaphragms. It suggests that alterations in calcium homeostasis are accumulating and some of these might be compensatory. This aspect is missing from the Discussion.

Fig 1. Colours used could be more distinct (especially the red spectrum) to make the identification of changed protein classes easier.

Fig 2. The use of, what appear to be, gene symbols in the bar diagram in addition to the protein names that were actually analyzed, is unclear. In any case, if these are gene names, the proper nomenclature for mouse genes should be applied, e.g., Dag1 not DAG1.

Author Response

Dear Reviewer 1,

Thank you very much for reviewing our manuscript [life-1960372] entitled ‘Proteomic identification of markers of membrane repair, regeneration and fibrosis in the aged and dystrophic diaphragm’ and your constructive criticism.

Please find below a point-by-point repsone to your individual comments: 

Reviewer 1, Comment 1: ‘Identification of disease biomarkers can help evaluation of changes that result from the progressing pathology, are adaptive and compensatory, or result from the therapeutic intervention. Therefore, this study contributes to this goal. Proteomic identification of changes in the expression levels of proteins involved in the build-up of extracellular matrix and ultimately fibrosis is interesting and these have the potential to be used as biomarkers in pre-clinical and, upon confirmation in DMD patients, clinical therapeutic studies’.

Response: We would like to thank Reviewer 1 for the overall positive evaluation of our submission and the constructive criticism of our report.

Reviewer 1, Comment 2: ‘However, the study is focused entirely on the abnormalities in dystrophic myofibres and seems to be leaving aside changes that can occur in myogenic cells. Given that the progressive loss of the regenerative capacity of dystrophic muscles leaves ultimately to patients’ deaths, it would be interesting to re-evaluate the proteomic data for biomarkers of muscle repair’.

Response: We would like to thank Reviewer 1 for this important point. Presentation of new data from the proteomic analysis has identified the potential myogenic markers cadherin-13 (T-cadherin) and CD34 in the data sets and shown increased levels in young and old mdx-4cv diaphragm muscle preparations. Since T-cadherin interacts with integrin and is linked to signal transduction proteins within caveolae structures, we have also added proteomic information on expression levels of  caveolin-1, caveolin-2, caveolin-3, integrin alpha-7 and integrin beta-1 to the revised manuscript. To better present theses additional results on myogenic markers and membrane repair markers, these findings were combined with the dysferlin data (from original Figure 3) and combined with the annexin diagram in new and revised Figure 6.
It is important to stress that our comparative proteomic study has used total extracts of skeletal muscle homogenates to evaluate major changes in the dystrophic diaphragm muscle during aging. The mass spectrometric analysis of crude extracts decisively reduces the potential introduction of artefacts that can be associated with extensive subcellular fractionation protocols. However, a bioanalytical disadvantage of working with total tissue extracts is the reduced coverage of low-abundance proteins that are for example present in cell types with a lower density as compared to bulk myofibres. Such proteoforms might be only identified by 1 peptide or are not present in multi-consensus files. In our experience, a comprehensive mass spectrometric coverage of myogenic regulatory factors requires considerable enrichment steps. Previous studies in our laboratory with affinity purified fractions have identified increased levels of the repair proteins myoferlin and dysferlin and this was confirmed by immunofluorescence microscopy and immunoblotting, respectively (Reference [112] Murphy et al. Proteomic analysis of the sarcolemma-enriched fraction from dystrophic mdx-4cv skeletal muscle. J. Proteomics 2019, 191, 212-227). Besides the mass spectrometric coverage of peptides generated from dysferlin, cadherin-13, integrins, caveolins and CD34, the suggested analysis of our data sets used in the current study did not result in the identification of additional repair proteins, transcription factors or myogenic markers. 

The new results on dysferlin, caveolins, integrins, cadherin-13 and CD34 have been integrated in the revised Introduction, Results and Discussion sections:

Revised Introduction section (line 90f): ‘… and reactive myofibrosis, i.e. annexins [58-60], dysferlin [61-63], caveolins [64-66], integrins [67-69], cadherin [70-72], CD34 [73-75], periostin [76-78] and collagens [79-81]. Changes in annexin 2, periostin and collagen VI were independently confirmed by comparative immunoblot analysis’.

Revised Results section (line 308f): ‘… Section 3.5. Proteomics of membrane repair and calcium sensing in mdx-4cv diaphragm. A protein of central importance in the calcium-dependent membrane repair process of damaged skeletal muscles is dysferlin (DYSF). The mass spectrometric survey of wild type versus mdx-4cv diaphragm preparations revealed an increase of this sarcolemmal repair protein. Since the dysferlin/myoferlin system closely interacts with caveolins and annexins, the expression levels of these crucial components of the caveolae structures and calcium regulation, respectively, were also examined. The concentration of caveolin 1 (CAV1) and caveolin 2 (CAV2) increased drastically in aged and dystrophic diaphragm, while the muscle-specific caveolin-3 (CAV3) isoform showed elevated levels in 3-month old mdx-4cv diaphragm but only marginal changes in 15-month old dystrophic muscles. Integrins are useful markers of proliferation and the main type present in the skeletal muscle periphery, a7b1-integrin (ITGA7/ITGB1), was identified to be increased in both young and aged mdx-4cv diaphragm. Interestingly, the expression of the myogenic markers cadherin-13 (CDH13) and CD34 were both elevated in muscular dystrophy. The calcium-dependent annexin isoforms …’.

Revised Discussion section (line 410f): ‘…and reactive myofibrosis. Annexin isoform ANXA2 [58-60], the matricellular protein periostin [76-78] and collagen isoform COL-VI [79-81], as well as the caveolins CAV1 and CAV2 [64-66], a7b1-integrin [67-69], cadherin-13/T-cadherin [70-72] and CD34 [73-75] were shown to be drastically increased in aged and dystrophin-lacking muscle preparations. Interestingly, the muscle-specific caveolin isoform CAV3 was only elevated in young and dystrophic diaphragm. Immunoblotting clearly verified the increased concentration of specific isoforms of annexin, periostin and collagen in muscular dystrophy. This suggests that these …’.

Revised Discussion section (line 544f): ‘… for muscle fiber membrane repair [58-62]. Dysferlin, myoferlin and caveolins closely interact in dystrophic fibres [45,111]. This makes both the ANXA2 isoform of muscle annexin and elevated levels of dysferlin in damaged and dystrophin-deficient fibers [112,131] promising biomarker candidates of calcium-dependent and dysferlin/myoferlin-associated repair processes in X-linked muscular dystrophy. Another interesting proteomic observation was the increase in integrins [67-69]. The a7b1-integrin complex is developmentally regulated and important for sarcolemmal stability and prevention of exercise-induced injury [132] and increased levels of a7b1-integrin were previously shown to counteract muscle degeneration [133-135]. Integrins play a key role in the reinforcement of crucial load-bearing structures at costameres and myotendious junctions. Together with the dystrophin-complex, integrins provide structural integrity for lateral and longitudinal force transmission across the sarcolemma [136]. Thus, the increased expression of a7b1-integrin might be part of the repair response in muscular dystrophy and represents a potential marker of cellular proliferation. Since this study was carried out with total extracts from crude muscle preparations, low-abundance markers were not detected. However, increases in the myogenic marker molecules CD34 [73-75] and cadherin-13 [71,72,137] were identified in the dystrophic mdx-4cv diaphragm by mass spectrometric analysis. The surface marker CD34 was recently shown to exhibit considerable potential as a satellite cell-linked biomarker of skeletal muscle aging [75]. Cadherin-13 (CDH13), also named heart cadherin (H-cadherin) or truncated cadherin (T-cadherin), attaches to the plasma membrane via a glycosylphosphatidylinositol anchor. Importantly, T-cadherin interacts with integrin-a7 and is linked to signal transduction proteins within caveolae structures [70]. This makes the identified alterations in the abundance of specific isoforms of integrin, cadherin and caveolin in muscular dystrophy an interesting finding with potential for the establishment of a comprehensive biomarker signature of dys-trophinopathy’.

Additional references [61]-[75] and [131]-[137] were added to the revised Introduction and Discussion sections of the R1version of our manuscript to cover previous work on dysferlin, caveolins, integrins, cadherin and CD34: 

[61] Lennon, N.J.; Kho, A.; Bacskai, B.J.; Perlmutter, S.L.; Hyman, B.T.; Brown, R.H. Jr. Dysferlin interacts with annexins A1 and A2 and mediates sarcolemmal wound-healing. J. Biol. Chem. 2003, 278, 50466-50473.
[62] Han, R.; Campbell, K.P. Dysferlin and muscle membrane repair. Curr. Opin. Cell Biol. 2007, 19, 409-416.
[63] Demonbreun, A.R.; Rossi, A.E.; Alvarez, M.G.; Swanson, K.E.; Deveaux, H.K.; Earley, J.U.; Hadhazy, M.; Vohra, R.; Walter, G.A.; Pytel, P.; McNally, E.M. Dysferlin and myoferlin regulate transverse tubule formation and glycerol sensitivity. Am. J. Pathol. 2014, 184, 248-259.
[64] Couchoux, H.; Bichraoui, H.; Chouabe, C.; Altafaj, X.; Bonvallet, R.; Allard, B.; Ronjat, M.; Berthier, C. Caveolin-3 is a direct molecular partner of the Cav1.1 subunit of the skeletal muscle L-type calcium channel. Int. J. Biochem. Cell Biol. 2011, 43, 713-720.
[65] Pradhan, B.S.; PrószyÅ„ski, T.J. A Role for Caveolin-3 in the Pathogenesis of Muscular Dystrophies. Int. J. Mol. Sci. 2020, 21, 8736.
[66] Matsunobe, M.; Motohashi, N.; Aoki, E.; Tominari, T.; Inada, M.; Aoki, Y. Caveolin-3 regulates the activity of Ca2+/calmodulin-dependent protein kinase II in C2C12 cells. Am. J. Physiol. Cell Physiol. 2022, 323, C1137-C1148.
[67] Collo, G.; Starr, L.; Quaranta, V. A new isoform of the laminin receptor integrin alpha 7 beta 1 is developmentally regulated in skeletal muscle. J. Biol. Chem. 1993, 268, 19019-19024.
[68] Burkin, D.J.; Wallace, G.Q.; Milner, D.J.; Chaney, E.J.; Mulligan, J.A.; Kaufman, S.J. Transgenic expression of {alpha}7{beta}1 integrin maintains muscle integrity, increases regenerative capacity, promotes hypertrophy, and reduces cardiomyopathy in dystrophic mice. Am. J. Pathol. 2005, 166, 253-263.
[69] Liu, J.; Burkin, D.J.; Kaufman, S.J. Increasing alpha 7 beta 1-integrin promotes muscle cell proliferation, adhesion, and resistance to apoptosis without changing gene expression. Am. J. Physiol. Cell Physiol. 2008, 294, C627-640.
[70] Philippova, M.P.; Bochkov, V.N.; Stambolsky, D.V.; Tkachuk, V.A.; Resink, T.J. T-cadherin and signal-transducing molecules co-localize in caveolin-rich membrane domains of vascular smooth muscle cells. FEBS Lett. 1998, 429, 207-210.
[71] Tanaka, Y.; Kita, S.; Nishizawa, H.; Fukuda, S.; Fujishima, Y.; Obata, Y.; Nagao, H.; Masuda, S.; Nakamura, Y.; Shimizu, Y.; Mineo, R.; Natsukawa, T.; Funahashi, T.; Ranscht, B.; Fukada, S.I.; Maeda, N.; Shimomura, I. Adiponectin promotes muscle regeneration through binding to T-cadherin. Sci. Rep. 2019, 9, 16.
[72] Nalbandian, M.; Zhao, M.; Sasaki-Honda, M.; Jonouchi, T.; Lucena-Cacace, A.; Mizusawa, T.; Yasuda, M.; Yoshida, Y.; Hotta, A.; Sakurai, H. Characterization of hiPSC-Derived Muscle Progenitors Reveals Distinctive Markers for Myogenic Cell Purification Toward Cell Therapy. Stem Cell Reports. 2021, 16, 883-898.
[73] Beauchamp, J.R.; Heslop, L.; Yu, D.S.; Tajbakhsh, S.; Kelly, R.G.; Wernig, A.; Buckingham, M.E.; Partridge, T.A.; Zammit, P.S. Expression of CD34 and Myf5 defines the majority of quiescent adult skeletal muscle satellite cells. J. Cell Biol. 2000, 151, 1221-1234.
[74] Jankowski, R.J.; Deasy, B.M.; Cao, B.; Gates, C.; Huard, J. The role of CD34 expression and cellular fusion in the regeneration capacity of myogenic progenitor cells. J. Cell Sci. 2002, 115, 4361-4374.
[75] Fernández-Lázaro, D.; Garrosa, E.; Seco-Calvo, J.; Garrosa, M. Potential Satellite Cell-Linked Biomarkers in Aging Skeletal Muscle Tissue: Proteomics and Proteogenomics to Monitor Sarcopenia. Proteomes. 2022, 10, 29.

[131] Vontzalidis, A.; Terzis, G.; Manta, P. Increased dysferlin expression in Duchenne muscular dystrophy. Anal. Quant. Cytopathol. Histpathol. 2014, 36, 15-22.
[132] Boppart, M.D.; Mahmassani, Z.S. Integrin signaling: linking mechanical stimulation to skeletal muscle hypertrophy. Am. J. Physiol. Cell Physiol. 2019, 317, C629-C641.
[133] Burkin, D.J.; Wallace, G.Q.; Nicol, K.J.; Kaufman, D.J.; Kaufman, S.J. Enhanced expression of the alpha 7 beta 1 integrin reduces muscular dystrophy and restores viability in dystrophic mice. J. Cell Biol. 2001, 152, 1207-1218.
[134] Rooney, J.E.; Gurpur, P.B.; Burkin, D.J. Laminin-111 protein therapy prevents muscle disease in the mdx mouse model for Duchenne muscular dystrophy. Proc. Natl. Acad. Sci. USA. 2009, 106, 7991-7996.
[135] Heller, K.N.; Montgomery, C.L.; Janssen, P.M.; Clark, K.R.; Mendell, J.R.; Rodino-Klapac, L.R. AAV-mediated overexpression of human a7 integrin leads to histological and functional improvement in dystrophic mice. Mol. Ther. 2013, 21, 520-525.
[136] Marshall, J.L.; Oh, J.; Chou, E.; Lee, J.A.; Holmberg, J.; Burkin, D.J.; Crosbie-Watson, R.H. Sarcospan integration into laminin-binding adhesion complexes that ameliorate muscular dystrophy requires utrophin and α7 integrin. Hum. Mol. Genet. 2015, 24, 2011-2022.
[137] Kim, H.; Perlingeiro, R.C.R. Generation of human myogenic progenitors from pluripotent stem cells for in vivo regeneration. Cell. Mol. Life Sci. 2022, 79, 406.

The newly introduced findings were taken into account in revised Figure 9 of the Discussion section. Reference numbers were changed accordingly.

Reviewer 1, Comment 3: ‘In this context, periostin, albeit it has a variety of functions in tissues, in DMD can hardly be considered a tissue regeneration marker. Its ablation alleviated the dystrophic phenotype (Lorts A, Schwanekamp JA, Baudino TA, McNally EM, Molkentin JD. Deletion of periostin reduces muscular dystrophy and fibrosis in mice by modulating the transforming growth factor-β pathway. Proc Natl Acad Sci U S A. 2012 Jul 3;109(27):10978-83.)’.

Response: To address this issue, we have revised our discussion of the multi-functional periostin protein and also modified the summarizing figure (now revised Figure 9) in the Discussion section. We have moved periostin to the section with ECM markers and list instead cadherin-13 and CD34 as increased myogenic markers. The manuscript by Lorts et al. is now discussed and has been added as new reference [130] in the revised manuscript.

Revised line 227f: ‘… reactive myofibrosis, multi-functional periostin (POSTN) as a prototype of a matricellular component and the membrane …’.

Revised line 509f: ‘… changes in the extracellular matrix [91,93,129]. Of note, the deletion of periostin was shown to have a positive effect on X-linked muscular dystrophy by reducing myofibrosis via modulation of the signalling pathway that is associated with transforming growth factor TFG-b [130]. Immunoblotting indicates that periostin exists only …’.

Revised line 512f: ‘This makes this component of the extracellular matrix an excellent candidate for evaluating progressive alterations in dystrophin-deficient skeletal muscles’.

Revised line 575f: ‘… and the matricellular protein periostin which plays a crucial role in the extracellular matrix, as well as various isoforms …’.

New reference [130]: Lorts, A.; Schwanekamp, J.A.; Baudino, T.A.; McNally, E.M.; Molkentin, J.D. Deletion of periostin reduces muscular dystrophy and fibrosis in mice by modulating the transforming growth factor-β pathway. Proc. Natl. Acad. Sci. USA. 2012, 109, 10978-10983.

Reviewer 1, Comment 4: ‘Given that the mdx mouse is the most widely used pre-clinical model of DMD, it would be important to test the key protein alterations in this mouse to confirm that the same changes occur in this natural mutant as in this chemically-induced mdx4cv mice used here’.

Response: We agree with this point and since we have previously carried out proteomic surveys of the mdx mutant, we are able to add discussions of findings from proteomics studies that have used the spontaneous and most widely employed mouse model of Duchenne muscular dystrophy. These investigations used different subtypes of skeletal muscles and differently aged groups of wild type versus dystrophic animals.

Revised Discussion on line 419f: “… genetic animal models of dystrophinopathy. Since the spontaneous mdx-23 mouse is the most frequently used animal model in muscular dystrophy research [47], it was of interest to compare the proteomic changes in the chemically induced mdx-4cv mouse [106-108] to the naturally occurring mdx-23 mutant [100-102]. The proteome-wide changes identified in this report on established markers of X-linked muscular dystrophy, i.e. decreases in all members of the core dystrophin complex and concomitant increases in vimentin, tubulin, ferritin, various molecular chaperones and lamin, and decreases in parvalbumin and carbonic anhydrase isoform CA3, agree with previous studies of the dystrophic mdx-23 mouse model [41-46,126,127]. An increased abundance of collagen COL-VI was also observed in mdx-23 muscle preparations using two-dimensional gel electrophoresis combined with staining by the fluorescent dye ruthenium II tris bathophenanthroline disulfonate [128]. New biomarker candidates can now be …”.

Reviewer 1, Comment 5: ‘It is interesting that most of the alterations in the calcium-related proteins were identified in old diaphragms. It suggests that alterations in calcium homeostasis are accumulating and some of these might be compensatory. This aspect is missing from the Discussion’.

Response: We agree that the observed proteome-wide changes in proteins involved in calcium homeostasis and the regulation of excitation-contraction coupling might represent accumulating abnormalities and/or compensatory mechanisms in dystrophic diaphragm fibres. This aspect is now discussed in the revised text.

Revised line 536f: ‘… decreased calcium buffering and abnormal calcium re-uptake into the lumen of the sarcoplasmic reticulum. These changes in ion-regulatory proteins might represent accumulating abnormalities and/or compensatory mechanisms in dystrophic diaphragm fibers [29]. Another interesting detection of …’.

Reviewer 1, Comment 6: ‘Fig 1. Colours used could be more distinct (especially the red spectrum) to make the identification of changed protein classes easier’.

Response: We would like to thank Reviewer 1 for pointing out this problem with the display of the pie chart. Since Reviewer 2 also commented on this issue with the displayed colour range in original Figure 1 (see: Reviewer 2, Comment 4), we have revised Figure 1 and now use directly descriptors of the different protein classes displayed in the pie charts. Since the colour coding of the output files from the bioinformatic PANTHER programme can unfortunately not be changed, we have instead split original Figure 1 into new Figures 1 and 2 in the revised R1 version of our manuscript. The name of an individual protein class is now directly attached to a specific part of the pie chart diagram in revised Figure 1 and new Figure 2. This should address any potential confusion about the pie chart display.

Revised text on lines 232 and 236 in Section 3.1.: ‘… by bioinformatic PANTHER analysis [99]. The findings are displayed in Figures 1 and 2, which show drastic alterations in a variety of protein types, including …’ and ‘… of metabolite interconversion enzymes during aging (Figure 2) ...’.
Revised figure legends of Figure 1 and new Figure 2:
Figure 1. Bioinformatic analysis of changed protein classes in 3-month old mdx-4cv diaphragm muscle. The analysis was carried out with the PANTHER program [99].
Figure 2. Bioinformatic analysis of changed protein classes in 15-month old mdx-4cv diaphragm muscle. The analysis was carried out with the PANTHER program [99].

Reviewer 1, Comment 7: ‘Fig 2. The use of, what appear to be, gene symbols in the bar diagram in addition to the protein names that were actually analyzed, is unclear. In any case, if these are gene names, the proper nomenclature for mouse genes should be applied, e.g., Dag1 not DAG1’.

Response: We would like to thank Reviewer 1 for highlighting this potential confusion in the description of the identified proteins. This point was also raised by Reviewer 2 (see: Reviewer 2, Comments 5-7). We do not use gene names in this manuscript on proteomic findings. In most proteomics publications, gene names are written in italics in the general text and the corresponding abbreviations of proteins/proteoforms are not; so that gene names are not mixed-up with protein names. We only once mention the ‘DMD’ gene in the Introduction section and it is written in italics. The rest of the manuscript does not refer to genes or gene names. 
In order to be consistent throughout the text, as well as figure legends and figures, and since this is a proteomic study, we now clearly describe in the text that all abbreviations of proteins are in capital and that they relate to protein names and not genes. We now use exclusively capital descriptors as abbreviations of protein names, so that they are not mixed-up with gene names.

Revised results section (line 221f): ‘… preparations, respectively. Throughout the manuscript, names of proteins, protein subunits or protein isoforms that were identified by proteomics are abbreviated in capital letters. Differential protein expression …’.

For example, the description of collagens now clearly lists the abbreviated names of collagen protein chains in capital letters.

Line 341f: ‘… increases in collagen I (COL1A1 and COL1A2 chains), collagen IV (COL4A1, COL4A2 and COL4A3 chains), collagen V (COL5A1 and COL5A2 chains), collagen VI (COL6A1, COL6A2, COL6A5 and COL6A6 chains), collagen XII (COL12A1 chain), collagen XIV (COL14A chain), collagen XV (COL15A1 chain) and collagen XVIII (COL18A1 chain), as shown in Figure 7. Increased …’.

Reviewer 2 Report

The Authors describe a comparative proteomic analysis of young versus aged diaphragm muscles from wild type versus the dystrophic mdx-4cv mice, to identify novel disease marker candidates for Duchenne Muscular Dystrophy.

The mass spectrometry-based proteomic screening of young versus aged mdx-4cv diaphragm muscle revealed complex proteome-wide changes in relation to dystrophin deficiency. Especially the Authors identified age-related increases in crucial markers of membrane repair regulation, tissue regeneration and reactive myofibrosis, i.e. annexin isoform ANXA2, the matricellular protein periostin, and collagen isoform COL-VI, independently confirmed by comparative immunoblot analysis.

The aim of the project is well defined, the results are clearly explained and the conclusions consistent with the results.

Minor revisions:

  • Line 64: please change “dysfunction” with “weakness”
  • Lines 63-68: please rephrase the sentence, highlighting that the pathology is mainly characterized by progressive muscle weakness, cognitive impairment in 30% of the DMD patients, articular deformities, scoliosis, and the occurrence of respiratory insufficiency and hearth failure which represent the main causes of death [ref: Duan D, Goemans N, Takeda S, Mercuri E, Aartsma-Rus A. Duchenne muscular dystrophy. Nat Rev Dis Primers. 2021 Feb 18;7(1):13. doi: 10.1038/s41572-021-00248-3. PMID: 33602943]. Then, I would mention minor complications as liver metabolism, the gastrointestinal tract, the immune system, kidney function.
  • Figure 1: color coding is not clearly readable: if possible, it would be appreciated adding more colors to make differences in purple and red-based colors more readable.
  • Lines 254-257: name of proteins are all in lowercase letters, while the same are in capital letters in the Figure 2: please make the formatting homogeneous between the Figure and the Figure legend.
  • Lines 271-274: same comment as per lines 254-257, for Figure 3.
  • Line 338: same comment as per lines 254-257, for Figure 6 (please change in capital letter collagen COL-VI, periostin and annexin ANXA2 to homogenize the Figure legend with the Figure)
  • Line 354: it is not clear what the terms “hypotrophy, hypertrophy and hyperplasia” are referred to. If they refer to “muscular”, please specify.
  • Discussion: at lines 229-231, the Authors state that “An interesting change between 3-month and 15-month old mdx-4cv diaphragm muscle is the drastic increase in the changed density of metabolite interconversion enzymes during aging”. This point is not further discussed. Can the Authors comment on that in the discussion?

Author Response

Dear Reviewer 2,

Thank you very much for reviewing our manuscript [life-1960372] entitled ‘Proteomic identification of markers of membrane repair, regeneration and fibrosis in the aged and dystrophic diaphragm’ and your constructive criticism.

Please find below a point-by-point respone to your individual comments:

Reviewer 2, Comment 1: ‘The Authors describe a comparative proteomic analysis of young versus aged diaphragm muscles from wild type versus the dystrophic mdx-4cv mice, to identify novel disease marker candidates for Duchenne Muscular Dystrophy. … The mass spectrometry-based proteomic screening of young versus aged mdx-4cv diaphragm muscle revealed complex proteome-wide changes in relation to dystrophin deficiency. Especially the Authors identified age-related increases in crucial markers of membrane repair regulation, tissue regeneration and reactive myofibrosis, i.e. annexin isoform ANXA2, the matricellular protein periostin, and collagen isoform COL-VI, independently confirmed by comparative immunoblot analysis. … The aim of the project is well defined, the results are clearly explained and the conclusions consistent with the results’.

Response: We would like to thank Reviewer 2 for the positive evaluation of our manuscript and the useful points made to improve the manuscript.

Reviewer 2, Comment 2: ‘Minor revisions: Line 64: please change “dysfunction” with “weakness”.’.

Response: This change was made in the revised manuscript.

Revised Introduction (line 63): “… are mainly characterized by progressive skeletal muscle weakness …’.

Reviewer 2, Comment 3: ‘Lines 63-68: please rephrase the sentence, highlighting that the pathology is mainly characterized by progressive muscle weakness, cognitive impairment in 30% of the DMD patients, articular deformities, scoliosis, and the occurrence of respiratory insufficiency and hearth failure which represent the main causes of death [ref: Duan D, Goemans N, Takeda S, Mercuri E, Aartsma-Rus A. Duchenne muscular dystrophy. Nat Rev Dis Primers. 2021 Feb 18;7(1):13. doi: 10.1038/s41572-021-00248-3. PMID: 33602943]. Then, I would mention minor complications as liver metabolism, the gastrointestinal tract, the immune system, kidney function’.

Response: We agree and have changed this part of the Introduction section accordingly.

Revised Introduction section (line 62f): “… in muscular dystrophy [27-29]. Dystrophinopathies are mainly characterized by progressive skeletal muscle weakness, cognitive impairment in a subset of Duchenne patients, articular deformities, scoliosis and the occurrence of cardio-respiratory failure, which represent the main cause of death [6,30,31]. In addition to skeletal muscle weakness, Duchenne muscular dystrophy is associated with a variety of less severe and body-wide complications affecting liver metabolism, the gas-trointestinal tract, the immune system and kidney function [30-32]. This makes X-linked muscular dystrophy …’.

Reviewer 2, Comment 4: ‘Figure 1: color coding is not clearly readable: if possible, it would be appreciated adding more colors to make differences in purple and red-based colors more readable’.

Response: We would like to thank Reviewer 2 for pointing out this problem with the display of the pie chart. Since Reviewer 1 also commented on this issue with the displayed colour range in original Figure 1 (see: Reviewer 1, Comment 6), we have revised Figure 1 and now use directly descriptors of the different protein classes displayed in the pie charts. Since the colour coding of the output files from the bioinformatic PANTHER programme can unfortunately not be changed, we have instead split original Figure 1 into new Figures 1 and 2 in the revised R1-version of our manuscript. The name of an individual protein class is now directly attached to a specific part of the pie chart diagram in revised Figure 1 and new Figure 2. This should address any potential confusion about the pie chart display.

Revised text on lines 232 and 236 in Section 3.1.: ‘… by bioinformatic PANTHER analysis [99]. The findings are displayed in Figures 1 and 2, which show drastic alterations in a variety of protein types, including …’ and ‘… of metabolite interconversion enzymes during aging (Figure 2) ...’.
Revised figure legends of Figure 1 and new Figure 2:
Figure 1. Bioinformatic analysis of changed protein classes in 3-month old mdx-4cv diaphragm muscle. The analysis was carried out with the PANTHER program [99].
Figure 2. Bioinformatic analysis of changed protein classes in 15-month old mdx-4cv diaphragm muscle. The analysis was carried out with the PANTHER program [99].

Reviewer 2, Comments 5-7: ‘Lines 254-257: name of proteins are all in lowercase letters, while the same are in capital letters in the Figure 2: please make the formatting homogeneous between the Figure and the Figure legend’. ‘Lines 271-274: same comment as per lines 254-257, for Figure 3’. ‘Line 338: same comment as per lines 254-257, for Figure 6 (please change in capital letter collagen COL-VI, periostin and annexin ANXA2 to homogenize the Figure legend with the Figure)’.

Response: To avoid any inconsistencies in protein names and their abbreviations, the revised manuscript uses in the figure legends and figures the full name of proteins in the same way and abbreviations are displayed in capital letters for the description of mass spectrometrically identified proteins. So for example in Figure 3 the protein Dystrobrevin is written as ‘a-Dystrobrevin’ and its abbreviation in capital letters as ‘DTNA’. 

Reviewer 2, Comment 8: ‘Line 354: it is not clear what the terms “hypotrophy, hypertrophy and hyperplasia” are referred to. If they refer to “muscular”, please specify’.

Response: The results outlined in reference [102] refer to muscular changes in the mdx-23 model of Duchenne muscular dystrophy. We have added this information to the revised Discussion section.

Revised Discussion section (line 386): ‘… This results in the almost complete deficiency of dystrophin isoform Dp427-M [47] and causes an X-linked myopathy in association with muscular hypotrophy, hypertrophy and hyperplasia in mdx-23 fibres [102]’.

Reviewer 2, Comment 9: ‘Discussion: at lines 229-231, the Authors state that “An interesting change between 3-month and 15-month old mdx-4cv diaphragm muscle is the drastic increase in the changed density of metabolite interconversion enzymes during aging”. This point is not further discussed. Can the Authors comment on that in the discussion?’.

Response: We would like to thank Reviewer 2 for pointing out this omission of discussing the findings of the PANTHER analysis. This has now been added to revised Discussion.

Revised Discussion section (line 447f): “… In relation to bioenergetic and metabolic enzymes, the bioinformatic PANTHER analysis of changed proteins revealed an extensive increase in the altered abundance of metabolite interconversion enzymes during aging of the dystrophic mdx-4cv diaphragm muscle. These considerable changes in the expression levels of muscle-associated enzymes included large numbers of hydrolases, isomerases, ligases, lyases, oxidoreductases and transferases. Within the protein class of hydrolases, this included proteins with amylase, deaminase, esterase, glucosidase, lipase, phosphatase, phosphodiesterase and pyrophosphatase activity. Isomerases were represented by epimerases, racemases and mutases. Lyases included aldolases, cyclases, dehydratases and hydratases. Alterations in the abundance of oxidoreductases encompassed dehydrogenases, oxidases, oxygenases, peroxidases and reductases. The protein family of transferases included acetyltransferases, acyltransferases, glycosyltransferases, kinases, methyltransferases and transketolases. A comprehensive biomarker signature of these types of tissue-associated changes in combination with … ’.

Round 2

Reviewer 1 Report

I am pleased that the authors considered the suggestions and revised the manuscript accordingly. I hope it resulted in an improved paper.